# (R)-Salbutamol Improves Imiquimod-Induced Psoriasis-Like Skin Dermatitis by Regulating the Th17/Tregs Balance and Glycerophospholipid Metabolism

**DOI:** 10.3390/cells9020511

**Published:** 2020-02-24

**Authors:** Fei Liu, Shanping Wang, Bo Liu, Yukun Wang, Wen Tan

**Affiliations:** 1Institute of Biomedical and Pharmaceutical Sciences, Guangdong University of Technology, Guangzhou 510006, China; 1111706007@mail2.gdut.edu.cn (F.L.); shanpingwang@outlook.com (S.W.); lb917752353@163.com (B.L.); Felixwyk@163.com (Y.W.); 2Jeffrey Cheah School of Medicine and Health Sciences, Monash University Malaysia, 47500 Bandar Sunway, Malaysia

**Keywords:** (*R*)-salbutamol, psoriasis, immune-regulation, Th17/Tregs, metabolomics

## Abstract

Psoriasis is a skin disease that is characterized by a high degree of inflammation caused by immune dysfunction. (*R*)-salbutamol is a bronchodilator for asthma and was reported to alleviate immune system reactions in several diseases. In this study, using imiquimod (IMQ)-induced mouse psoriasis-like dermatitis model, we evaluated the therapeutic effects of (*R*)-salbutamol in psoriasis in vivo, and explored the metabolic pathway involved. The results showed that, compared with IMQ group, (*R*)-salbutamol treatment significantly ameliorated psoriasis, reversed the suppressive effects of IMQ on differentiation, excessive keratinocyte proliferation, and infiltration of inflammatory cells. Enzyme-linked immunosorbent assays (ELISA) showed that (*R*)-salbutamol markedly reduced the plasma levels of IL-17. Cell analysis using flow cytometry showed that (*R*)-salbutamol decreased the proportion of CD4+ Th17+ T cells (Th17), whereas it increased the percentage of CD25+ Foxp3+ regulatory T cells (Tregs) in the spleens. (*R*)-salbutamol also reduced the increased weight ratio of spleen to body. Furthermore, untargeted metabolomics showed that (*R*)-salbutamol affected three metabolic pathways, including (i) arachidonic acid metabolism, (ii) sphingolipid metabolism, and (iii) glycerophospholipid metabolism. These results demonstrated that (*R*)-salbutamol can alleviate IMQ-induced psoriasis through regulating Th17/Tregs cell response and glycerophospholipid metabolism. It may provide a new use of (*R*)-salbutamol in the management of psoriasis.

## 1. Introduction

Psoriasis is a widespread chronic skin disorder with a 2–3% prevalence rate worldwide. The condition occurs as a result of an inflammatory reaction mediated by the immune system. It usually starts by chemokines and cytokines drawing cells of the immune system to the psoriatic area of the skin. This is quickly followed by invading and local cell proliferation [1,2,3]. However, psoriasis can also be caused by systemic low-grade inflammation, which may be associated with co-morbidities such as metabolic syndrome and cardiovascular disease [4,5]. Although studies are yet to reveal the precise cause of the condition, genotypic dependency, and environmental stimuli such as infection, stress, drugs and trauma have been implicated as possible causes of the diseases [6]. Psoriasis affects the skin’s appearance, therefore compromising the quality of patients’ life considerably [7]. Although several short-term and long-term treatment options for the disease exist, a complete cure is yet to be achieved. The disease was considered initially as an autoimmune disease mediated by Th1 [8]. However, accumulating evidence suggests that the cytokines interleukin (IL)-17, Foxp3 induced by Th17 or neutrophils and Treg cells, respectively, are liable for the increased prevalence of psoriatic dermatitis skin and could play an essential role in the disease development [9,10,11].

Metabonomics is a current high-throughput technique [12], which offers quantitative measures of biological systems involving global alterations in individuals’ metabolic profiles in response to genetic modification or pathophysiological stimuli [13]. Metabolic product profiling i.e., metabotyping or metabolic phenotyping has offered newfangled insights into the metabolic syndrome [14]. The metabonomics approach has gained application in diagnosis of diseases, evaluation of the efficacy, and toxicity of drugs and investigations on pathogenesis of diseases [15,16]. Using metabolomics, Ueharaguchi Y et al. elucidated the possible pro-psoriatic dermatitis mechanism of Thromboxane A2 [17]. Arnald Alonso et al. studied the mechanism of immune-mediated inflammatory diseases [18]. To date, however, metabolomics analysis in mouse psoriasis has been rarely performed.

The use of β_2_ adrenergic receptor agonists is a popular treatment option for obstructive diseases of the lungs and owns a number of anti-inflammatory effects [19,20]. Topical use of (*R*)-salbutamol cream (0.5%) exhibits effectiveness in treatment of sub-acute lupus erythematosus (SCLE) and discoid lupus erythematosus (DLE) [21]. However, the effects of (*R*)-salbutamol in treatment psoriasis has not been reported, and its effects on the metabolomics and biochemistry analysis in psoriasis has yet to be done.

Imiquimod (IMQ), which is a toll-like receptor 7/8 agonist, is a robust immune activator [22]. The topical application of IMQ in mice causes psoriasis-like skin, which is similar to human psoriasis dermatitis phenotypically and histologically [23]. The present study aimed to evaluate (*R*)-salbutamol effects on IMQ-induced skin dermatitis (psoriasis-like) in BALB/c mice. The effect of (*R*)-salbutamol on the regulation of Th17/Tregs cells and one profile of metabolomics were also investigated. This study provides a new use of (*R*)-salbutamol in treatment of psoriasis and the underlying mechanism.

## 2. Materials and Methods

### 2.1. Materials

Animal experiments were conducted using 8–11 weeks-old BALB/c female mice weighing 20–25 g (Certification No. 44007200057513, SPF grade) purchased from Guangdong Medical Laboratory Animal Center (Guangdong, China). This study was approved the Institutional Animal Care and Research Advisory Committee of the Institute of Biomedical and Pharmaceutical Sciences at Sun Yat-Sen University (permit number IACUC-2013-1204; approved 1 December 2013). Imiquimod cream (19040540) was obtained from Med-Shine Pharmaceutical Co., Ltd. (Sichuan, China) and (*R*)-salbutamol (>99% purity, 99.85% ee) was from Dongguan Key-Pharma Biomedical Co., Ltd. (Dongguan, China). ELISA kits of IL-17 (M181112-008a) was from Neobioscience, (Shenzhen, China). All of antibodies were acquired from Biolegend (San Diego, CA, USA).

### 2.2. Mice Treatments

The mice were divided into six groups (8 mice/group), i.e., imiquimod-treated group (IMQ), IMQ plus high dose of (*R*)-salbutamol group (H, 2 mg/kg), IMQ plus medium dose of (*R*)-salbutamol group (M, 1 mg/kg), IMQ plus low dose of (*R*)-salbutamol group (L, 0.5 mg/kg), IMQ plus dexamethasone group (Dex, 1 mg/kg) and control group (C). The mice were given water and normal forage for three days. An area of about 5 cm × 4 cm on the back skin of each mouse was shaved. In IMQ treatment, the mice received a daily topical application of 62.5 mg of 5% imiquimod cream for seven days. The control group received Vaseline. (*R*)-salbutamol and Dex were administered to mice orally (p.o.) twice a day for seven days. Mice in the IMQ group were administered with distilled water instead of the drugs. On the 8th day, mice were sacrificed to collect plasma, spleen and skins samples. 

### 2.3. Evaluating the Severity of Skin Inflammation

The degree of inflammation on the back skin was determined using the clinical Psoriasis Area and Severity Index (PASI) (Appendix A). Of note, we did not include the affected skin area in the calculation of overall scores. Scores ranging from 0 to 4 were used to rank the scaling, thickening, and erythema separately. In detail, scores 4, 3, 2, 1, and 0 were interpreted as very striking, moderate, slight, and none, respectively. The assessment of the morphological changes of dermatitis in mice was based on the PASI scoring system, with scores ranging from 0 to 12. Finally, skin dermatitis was evaluated using average scores for each group.

### 2.4. Histopathological Examination

After harvesting, the back skin specimen was fixed in 4% paraformaldehyde for 24 h prior to embedding in paraffin. Thereafter, they were cut into appropriate sections that were subjected to hematoxylin and eosin (HE) staining. The histopathological features of the sections were examined under a microscope (Nikon, Tokyo, Japan). The Baker scoring criteria were used to analyze the histopathological changes (Appendix A).

### 2.5. Hematological Analysis

The IDEXX ProCyte DX hematology analyzer (IDEXX, Westbrook, ME, USA) was employed for hematological analysis. This test was performed using whole blood samples preserved in EDTA tubes. The measurement of the white blood cells (WBC), neutrophil (NEUT) and mononuclear cells (MONO) were made in the hematology analyzer.

### 2.6. IL-17 ELISA

An enzyme-linked immunosorbent assay (ELISA) kit (Neobioscience, Shenzhen, China) was employed to measure the serum levels of Plasma IL-17 based on a standard curve created from serial dilutions of anti-IL-17 antibody.

### 2.7. Intracellular Staining and Flow Cytometry

Mice spleens were excised and bathed in PBS solution in a culture dish. The specimens were homogenized and filtered via a 70 µm nylon mesh. The tissue homogenate was mechanically dissociated to release single cells which were lysed in erythrocyte lysis buffer. Finally, they were suspended in PBS (without magnesium and calcium ions).

To perform intracellular staining, Splenocytes were harvested and incubated with magnetic-activated cell sorting buffer (MACS, 0.5% BSA and 2 mM EDTA in PBS) blocking with Fc receptors. This was followed by staining with the following antibodies: Alexa Fluor 488–anti-CD3 (B284975), BV605–anti-CD4 (B279163), BV510–anti-IL-17 (B263584) and PE-anti-Foxp3 (B275698). The cell activation cocktail (with Brefeldin A) containing phorbol myristate acetate (PMA, protein kinase C activator, 40.5 µM), ionomycin (Ca**^2+^** ionophore, 669.3 µM) and brefeldinA (Golgi inhibitor, 2.5 mg/mL) (B265097) was used to stimulate the splenocytes for 6 h. After incubation, we blocked the harvested splenocytes by treating them with MACS, followed by surface-staining, fixing, and permeabilization with the Fixation/Permeabilization Kit (B280105).

### 2.8. Metabolomic Analysis

Metabolite extraction of plasma: A three solvent biphasic system: water: methanol: methyl-T-butyl-ether at the volumetric ratio of 1:3:6 (MTBE solution) was used to extract metabolites in this study [24,25]. Briefly, 160 µL of MTBE solution was mixed with 40 µL of plasma and vortexed for 30 min at 4 °C. The mixture was centrifugated (3000× *g*, 30 min, 4 °C) which formed two fractions, (1) a hydrophilic fraction comprising water and methanol (2) and an organic hydrophobic fraction comprising methanol and MTBE. These fractions were vacuum-dried and then suspended in 45 µL 0.1% formic acid in water in preparation for analysis. The quality control (QC) test was performed using 600 µL from each sample to accurately represent the breadth of metabolites [26,27]. The blank was injected during the initial run to condition the column. The QC sample was injected as six replicates to increase the precision of injection. To test the stability of the system, a QC sample from every five experimental samples was analyzed.

UHPLC/ESI-TIMS TOF-MS/MS data acquisition and analysis: The UHPLC/ESI-TIMS TOF-MS/MS system in conjunction with the Trapped Ion Mobility spectrometry and TOF mass spectrometry (Bruker Daltonics Inc., Billerica, MA, USA) in positive and negative ion mode and the Dionex UltiMate 3000 RSLC system (Thermo Scientific/Dionex, Amsterdam, The Netherlands) equipped with an Acquity UPLC BEH-C18 column (2.1 mm × 50 mm, 1.7 µm) were used to analyze the samples. We acquired MS/MS data by combining the AutoMS/MS scanning experiment with data-dependent acquisition (DDA) model, which allowed the precursor ion to be selected as the most intense peak during LC–MS analyses. The samples used in this experiment were stored at 4 °C and analyzed using 5 µL.

Sample injection description: for each sample, we prepared two fractions from biphasic extractions [28], an aqueous layer and an organic layer. We first injected 5 µL of the organic layer, followed by an injection of 5 µL of the aqueous phase to the same column to prepare the aforementioned gradient. This step lasted for 1 min and the gradient was not initiated. Meanwhile, the concentration of acetonitrile (B solvent) in the mobile phase was not increased, this was done to ensure that the hydrophobic lipids were ahead in the column. Immediately following (via the next line in the sequence table), we injected 5 µL of the aqueous phase to the same column, after which a full gradient was initiated.

### 2.9. Statistical Analysis

The data of UHPLC/ESI-TIMS TOF-MSMS were processed for peak alignment, picking, and normalization using Progenesis QI software (Waters, Manchester, United Kingdom) to generate peak intensity for *m*/*z* data pairs and retention time (TR). The resultant data matrix was analyzed using EZinfo 3.0 software for PCA, PLS-DA, and OPLS-DA. We screened potential biomarkers that discriminate the IMQ/(R)-salbutamol and control group by metabolomics analysis. The max fold change >2 values *p* < 0.05 and variable importance in projection (VIP) values >1.0 were used as the cutoff values. The metabolite peaks were identified by MSMS analysis or by screening the biochemical databases, such as KEGG, Lipidmaps and HMDB. The OmicShare tools (www.omicshare.com/tools) were used to establish the Venn diagrams. The MetaboAnalyst 4.0 software (McGill University, Montreal, QC, Canada) and KEGG pathway database were employed for pathway analysis.

Other statistical analyses were performed as means ± standard deviation (SD). Data statistics were performed and presented by GraphPad prism-5 software (GraphPad Software Inc., La Jolla, CA, USA). The one-way ANOVA and *t* Student’s-test were used to compare differences between the IMQ and (*R*)-salbutamol plus imiquimod-induced groups. *p* < 0.05 was considered statistically significant.

## 3. Results

### 3.1. (R)-Salbutamol Alleviates Psoriatic Dermatiti

The effects of (*R*)-salbutamol on psoriasis was examined using imiquimod (IMQ) induced psoriasis mouse model, which is known to trigger and aggravate psoriasis both locally and in distance sites. Here, we explored whether (*R*)-salbutamol can prevent psoriasis. IMQ-was administered to mice to induce psoriasis-like skin inflammation. (*R*)-salbutamol treatment was administered at different doses as shown in Figure 1, given orally twice daily followed by treatment with IMQ at the shaved site of mouse back skin once daily for seven consecutive days (Figure 1). Typical psoriasis symptoms occurred in mice and manifested with lesions and severe inflammation reactions after topical application of IMQ. Oral administraion of (*R*)-salbutamol or Dexamethasone markedly alleivated the pathological changes induced by IMQ in a dose-dependent manner (Figure 2A). The PASI scoring system was employed to assess the severity of dermatitis according to erythema, skin thickening, scaling and calculative score as shown in Figure 2B–E. The scores were lower in mice treated with (*R*)-salbutamol or Dex than mice that received IMQ group (*p* < 0.01 or *p* < 0.05).

### 3.2. (R)-Salbutamol Alleviates the Pathology Changes Alterations Caused by IMQ on Mice Skin

The occurrence of psoriasis-like dermatitis following the application of IMQ was confirmed by HE examination, which showed the high inflammatory cell infiltration, acanthosis with extended rete ridges, Munro’s microabscesses, parakeratosis, high number of prickle cells, and thickened prickle cell layer of the epidermis. In contrast, mice treated with (*R*)-salbutamol (L, M and H) and Dex group, had mild epidermal thickening, parakeratosis and smoother epidermis (Figure 3A). Of note, microscopic examination showed that IMQ treatment increased the thickness of the epidermis increased by 3 to 4-fold relative to the control group. Interestingly, epidermal thickening in the (*R*)-salbutamol or Dex groups was significantly reduced (*p* < 0.01 or *p* < 0.05) (Figure 3B). Further analysis showed that mice treated with IMQ had higher Baker scores relative to the control group mice (Figure 3C), and those that received Dex and (*R*)-salbutamol (*p* < 0.05, *p* < 0.01), indicating that (*R*)-salbutamol effective against the psoriasis pathological changes induced by IMQ in mice.

### 3.3. Effect of (R)-Salbutamol on Haematological Parameters of IMQ-Induced Mouse Psoriasis 

To examine whether (*R*)-salbutamol suppresses inflammatory responses in the blood, we analyzed haematological parameters using an IDEXX ProCyte DX hematology analyzer. We observed that IMQ group had significantly increased total number of WBC, NEUT and MONO (*p* < 0.01) compared to the control group. Interestingly, oral administration of (*R*)-salbutamol significantly reduced the number of WBC, NEUT and MONO, This inhibitory effects of (*R*)-salbutamol was higher than Dex. This results demonstrated that (*R*)-salbutamol can significantly suppresses the inflammatory responses in the blood in psoriasis mice (Figure 4A–C).

### 3.4. (R)-Salbutamol Reduced IL-17 Secretion in Mice Plasma

The levels of IL-17 in the plasma was up-regulated in IMQ psoriasis mice. Compared with the control group, the level of IL-17 in the IMQ model group was increased by 5.82-fold, indicating an enhancement of immune inflammatory reaction. However, this increament of IL-17 in IMQ psoriasis mice was mostly diminished by (*R*)-salbutamol (*p* < 0.01) (Figure 4D).

### 3.5. Effect of (R)-Salbutamol Treatment on the Ratio of Spleen Weight to Body Weight 

The size and weight of the spleen were markedly enlarged in IMQ-induced psoriasis mice comparing to the control. The ratio of spleen to body weight was increased in IMQ psoriasis mice. However, the enlarged spleen was reduced by the treatment of (*R*)-salbutamol, and the ratio of spleen to body weight also significantly decreased by (*R*)-salbutamol in comparing to IMQ psoriasis mice. (Figure 5A,B). Dexamethoson had a similar effect as (*R*)-salbutamol.

### 3.6. (R)-Salbutamol Immune-Regulates the Number of CD3+CD4+ T Cells in Mice Treated with IMQ

Regulatory T cells (Treg) are known to normalized dysregulated autoimmune inflammatory reaction and prevent the autoimmune diseases, such us psoriasis [29]. It is also reported that IMQ application can modulate the distribution of Th cell types [23]. The influence of (*R*)-salbutamol on Th cells (Th17 and Treg) in mouse spleen were examined. After 7 days application of IMQ, mice were sacrified and spleens were taken. The isolated spleen cells were exposed to an activation cocktail for 6 h. Then, flow cytometry was conducted to quantify the amount of surface CD4, CD3, CD25 and the level of intracellular IL-17 and Foxp3. Results showed that treatment with IMQ markedly reduced the number of CD3+CD4+T cells (Figure 6A,B). Surprisingly, the inhibitions were significantly diminished by the treatment of (*R*)-salbutamol or Dexamethasone (Figure 6A,B). The different type of Th cells were also affected by (*R*)-salbutamol. The numbers of CD4+IL-17+ T cells (Th17) were increased after 7 days of application of IMQ (Figure 6C,E). However, this increment was totally diminished by (*R*)-salbutamol (Figure 6C,E). Dexamethasone had no similar effect on Th17 levels as (*R*)-salbutamol (Figure 6C,E). On the other hand, the number of CD25+Foxp3+ T cells (Tregs) were dereased in MIQ psoriasis mice comparing to control. The decreased level of Treg were restored to control by low dose of (*R*)-salbutamol and the level of Treg level were further increased above the control by high dose of (*R*)-salbutamol (Figure 6D,F). These data suggest that (*R*)-salbutamol antagonized the effects of IMQ by a differentiate in regulating Tregs and Th17 cells.

### 3.7. The Influence of (R)-Salbutamol on Metabolic Effects of IMQ Treatment

To investigate the effects of (*R*)-salbutamol on metabolic pathways and metabolite changes, an untargeted metabolomics approach was employed. Two sets of quality control and parallel samples were used to optimize the repeatability, precision and stability, of UHPLC-TIMSTOF-MS/MS methods. To validate the analytical method, retrieved ion chromatographic peaks of six ions in positive ion mode (with the *m*/*z* pairs and retention time of 0.40–203.0305, 2.50–406.6684, 4.00–437.8683, 6.00–550.9698, 12.35–496.2848, 13.60–524.3127) and six ions in negative ion mode (with the retention time and *m*/*z* pairs of 0.40–215.0183, 2.10–203.0688, 4.20–348.1693, 11.50–538.2785 13.30–568.3236, 15.80–281.2295) were selected. The RSDs of retention time for system stability, repeatability, and precision of injection were calculated as 0.00–2.24%, 0.00–0.44% and 0.00–0.44%; whereas the RSDs of peak area were 2.41–8.00%, 1.05–12.82% and 1.20–17.47% (Appendix A). These data indicated that the method is suitable for metabolomics analyses and has excellent repeatability and stability.

In the analysis of the separation between experimental groups, highly sensitive modeling methods e.g., Progenesis QI software combined with PCA and TIMS-TOFMS/MS analysis platform are often employed. The score plot of PCA in negative and positive ionization modes revealed that IMQ psoriasis group was differentially clustered among control and different doses of (*R*)-salbutamol groups, in the direction of the first principal component (R2X = 82% in negative mode, R2X = 84% in positive mode) (Figure 7A,B), indicating that the endogenous metabolite profiles of the control group were significantly different from that of IMQ group. In addition, the metabolite profiles of (*R*)-salbutamol and IMQ groups clustered separately, implying that the metabolites generated by IMQ are modulated by (*R*)-salbutamol.

Further, OPLS-DA and PLS-DA were employed to maximize the discrimination and identification of metabolites in the groups. Notably, the cumulative predictive capacity of cells treated with (*R*)-salbutamol was higher (Q2 = 0.85 in negative mode, Q2 = 0.81 in positive mode,) with good faithful representation of the data (R2Y = 0.96 in positive mode, R2Y = 0.95 in negative mode) (Figure 7C,D). These findings indicated (*R*)-salbutamol could alter the metabolome of IMQ-induced mouse psoriasis. Additionally, our data suggested that the metabolite profiles of the (*R*)-salbutamol (L, M, H) group were overlapped (Figure 7A–D), suggesting that different doses of (*R*)-salbutamol altered IMQ-induce psoriasis metabolites similarly.

Moreover, volcano plots revealed that the up-regulated (red plots) and down-regulated (green plots) metabolites were significantly different between the groups (Figure 7G,H). Additionally, a Venn diagram was created to compare the characteristic of each metabolite in IMQ-induced and (*R*)-salbutamol groups (Figure 7E,F). Based on fold change >2, variable importance (VIP) value >1 and *p* < 0.05, LipidMAPS, EZinfo software and HMDB analyses uncovered 39 biomarkers from the peak spectrum of the metabolomics. Metabolites such as *N*-Despyridinyl rosiglitazone, Dibucaine, Coutaric acid, PC (20:4(5Z,8Z,11Z,14Z)/0:0), PS (22:1(11Z)/0:0), PC (20:3(8Z,11Z,14Z)/0:0) were identified (Appendix A). Among those metabolites, glycerophospholipid metabolism was increased in all the IMQ-induced, and (*R*)-salbutamol groups. Metabolites, including PC(20:4(5Z,8Z,11Z,14Z)/0:0), PS(22:1(11Z)/0:0), PC(20:3(8Z,11Z,14Z)/0:0) were increased significantly in IMQ-induced group; whereas the metabolites were decreased significantly in (*R*)-salbutamol when compared to IMQ-induced group (Appendix A). The alterations in the level of biomarkers were visualized using a Heat map of unsupervised hierarchical clustering. The heat map (color changes from blue to red) indicated that those down-regulated and up-regulated metabolites in those groups (Figure 7I). Based on the color distribution, those cells treatments with (*R*)-salbutamol showed a closer trend to that control group when compared to the IMQ-induced group. Among all these biomarkers, the contents of *N*-Despyridinyl rosiglitazone, dibucaine, saxagliptin, PSF-A, 11Z-heptacosene, petasinoside, PC(20:3(8Z,11Z,14Z)/0:0), PI(20:4(5Z,8Z,11Z,14Z)/22:2(13Z,16Z)) in IMQ model group were significantly higher than those in control group, and most of them were decreased in (*R*)-salbutamol group (Figure 8). Besides, a similar trend was also found for the contents of some other metabolites in plasma samples (Appendix A), which suggested that (*R*)-salbutamol can alter the IMQ-induced psoriasis mouse plasma metabolic pattern with a higher efficacy.

MetaboAnalyst 4.0 was used for searching the possible metabolic pathways targeted using the candidate biomarkers in (*R*)-salbutamol treated IMQ-psoriasis mice. Some related pathways that likely involved in (*R*)-salbutamol mediated anti-psoriatic were identified in this study. The most significant pathways linked to the effects of (*R*)-salbutamol in IMQ-induced psoriasis were those of sphingolipid metabolism, arachidonic acid metabolism and glycerophospholipid metabolism (Figure 7J). The data presented here show that the observed variations stemmed from changing of endogenous metabolites among groups of control, IMQ-psoriasis and IMQ psoriasis treated with (*R*)-salbutamol mice.

## 4. Discussion

The current study evaluated the effects of commonly used 1 mg/kg (*R*)-salbutamol in treatment of psoriasis and in regulating Th17/Tregs cells response and metabolism in IMQ-induced psoriasis in mice. β_2_ receptor agonists are bronchodilators used in the treatment of obstructive pulmonary diseases [19]. It is reported that the (*R*)-salbutamol of β_2_ receptor agonists can also reduce inflammation of the respiratory tract, suppress proinflammatory mediators, thus preventing exudate production and tissue edema [30,31,32]. Racemic salbutamol relieves carrageenan-induced paw edema in rats via a β_2_ receptor-dependent mechanism [33]. An explorative clinic investigation indicated that (*R*)-salbutamol can treat discoid and subacute lupus erythematosus [21]. This study established a significant alleviation effects of (*R*)-salbutamol on IMQ-induced psoriasis-like skin dermatitis in mice. The anti-psoriatic effects of (*R*)-salbutamol involves in regulating the Th17/Tregs blance and glycerophospholipid metabolism.

Previous research has suggested that psoriasis could be caused through complex interactions between the immune system, autoantigens, several environmental factors, and susceptible loci associated with psoriasis [34,35,36]. InIMQ-induced mouse psoriasis. IMQ activates WBCs, neutrophils and monocyte cells through TLRs, initiating cytokine signaling cascade, and induce IL-17 expression and other cytokines [37]. Cellular infiltration represents an essential feature of skin inflammation, while WBCs, neutrophils, and monocytes are the main cell types that infiltrate the area [38]. These cells playan essential roles in the development of skin dermatitis. Herein, the application of imiquimod cream resulted in the formation of clear psoriasis-like pathological alterations. These observations agreed with a study conducted previously [39]. Our results showed that (*R*)-salbutamol reduced erythema, skin thickness, scaling, and confer protective effects against psoriasis dermatitis induced by IMQ. In addition, histopathological results confirmed that (*R*)-salbutamol prevented the proliferation and abnormal differentiation of IMQ-induced keratinocytes. Hematological evaluation revealed significant reduction in IMQ-stimulated increases in the number of WBCs, neutrophils and monocytes, which lead to a significant suppression in inflammatory responses Moreover, (*R*)-salbutamol reduced the number of neutrophils and the levels of IL-17 in mice plasma, the latter produced from either Th17 or neutrophils T cells [11].

Psoriasis is an autoimmune disease that is mediated by T cell dysfunction [40]. Th17 cells, Treg cells, and associated cytokines were thought to be vital for psoriasis development [41]. The Th17/Treg axis is crucial for the development of chronic disease and has been the focus of drug development [42]. The Treg-foxp3-driven disorder is regarded as a potentially more critical pathway in the initial stage of psoriasis [43,44]. In this study, IMQ application results in significantly increased levels of CD4+ Th17+ T cells (Th17) and decreased levels of CD3+CD4+ T cells and CD25+ Foxp3+Tregs in comparisons with control groups. These results are consistent with the findings of an earlier study [45]. In the IMQ induced psoriasis-like mice, (*R*)-Salbutamol administration reduced the expression of IL-17 in plasma along with a downregulation of the CD4+ Th17+ T cells (Th17) and upregulation of the CD3+CD4+ T cells and CD25+ Foxp3+ Tregs in the spleens. Collectively, these findings indicated that (*R*)-salbutamol attenuates psoriasis-like skin dermatitis induced by IMQ by regulating Th17 and Treg differentiation and inhibiting cytokines secretion. These results are not in agreement with reports in which treatment of racemic salbutamol regulates the balance between Th1 and Th2 cytokines instead of Th17 and Treg [46]. Additionally, Dex markedly reduces the levels of IL-17 in mice plasma (Figure 4D) and neutrophils (Figure 4C) in mice blood, suggesting dexamethasone alleviates psoriasis by reducing the production of IL-17 from neutrophils rather than from Th17 cells [46]. However, the inhibition effects of Dexamethasone on total WBC, neutrophil, and monocyte in blood were much more less significant in comparing (*R*)-salbutamol in IMQ induced Psoriasis mice.

The spleen is a major organ of the immune system that reflects the immune system of the human body and producing a variety of immune-active cytokines, hence it plays an important role in immune activities [47]. In this study, splenomegaly was induced by this systemic effects of topical IMQ application and with an increase of to body weight ratio. This is due to an increase in cellular proliferation in the spleen, activated by inflammatory immune responses and it was verified in the current study. This psoriasis-induced splenomegaly was greatly ameliorated by (*R*)-salbutamol, indicating systemic anti-psoriatic effects of (*R*)-salbutamol by regulating inflammatory immune cells production of spleen in inflammatory immunoreactions.

Previous studies showed that altered small molecules metabolism play key roles in the immune dysfunction in autoimmune diseases, including psoriasis, systemic lupus erythematosus, and rheumatoid arthritis [48,49]. In the untargeted metabolomics, employing multivariate statistical analysis, alterations in metabolites that are involved in the IMQ-induced psoriasis were identified. PCA and PLS-DA showed significant differences in the metabolic profiles among the (*R*)-salbutamol-treated (L, M, H), IMQ-induced, and control groups. The metabolic profiles of IMQ treated was significantly different in comparing to the control group, while the (*R*)-salbutamol treatment were very similar to that of the control group. These observations provided further evidences for the therapeutic effects of (*R*)-salbutamol against IMQ-induced psoriasis in mice. The untargeted metabolomics data of the present study identified 40 new potential biomarkers related to metabolic dynamics in IMQ-induced e psoriasis before and after treated with (*R*)-salbutamol. Glycerophospholipids, sphingolipid and arachidonic acid were the primary metabolites that affect IMQ induced psoriasis. These may be the major metabolic pathway that regulated the pathological process and the underlying molecular mechanism of this psoriasis. Glycerophospholipids are vital for maintain normal physiological cellular functions and as necessary as proteins and genes [50]. Its downregulation may contribute to cell membrane damage. Phosphatidylcholines (PC) and phosphatidylserine (PS) are the main glycerophospholipids found in the phospholipid membranes [50]. Recent studies revealed that the higher proliferation rates of skin cells could also explain the lower concentrations of PC in psoriasis plasma, which is an essential component of cell membranes [49]. Furthermore, reduced levels of PS were found in active psoriasis [51]. In our study, PC and PS had lower concentrations in the IMQ-induced groups, but which were up-regulated by (*R*)-salbutamoltreatment, indicating that (*R*)-salbutamol conferred anti-psoriatic effects by modulating the metabolism of glycerophospholipid. Several studies [51,52,53] have reported the importance of sphingolipids in innate immunity regulation, especially in T cells differentiation and programming. Our results discovered that (*R*)-salbutamol against IMQ induced psoriasis by regulates the Th17/Treg axis. The metabolomics results suggest that the regulation of Th17/Tregs by (*R*)-salbutamol may involve in sphingolipids. The biomarkers indentified in this study either in the IMQ-induced mouse psoriasis or after, (*R*)-salbutamol treatment could be useful in identify pathways and mechanism mediating the pathogenesis of psoriasis.

In summary, this study demonstrates a significant anti-psoriasis effects by (*R*)-salbutamol. This may involve regulateing the Th17/Tregs axis balances and glycerophospholipid metabolism in response to IMQ induced psoriasis (Figure 9), (*R*)-salbutamol also revealed a better therapeutic effect than corticosteroid in against IMQ-induced psoriasis in this study. (*R*)-salbutamol has been used years in clinic for anti-asthma and COPD with well accepted safety-profile. Therefore, the findings of the this study provide a better alternative for the unmet medical need in treatment of psoriasis.

## Figures and Tables

**Figure 1 cells-09-00511-f001:**
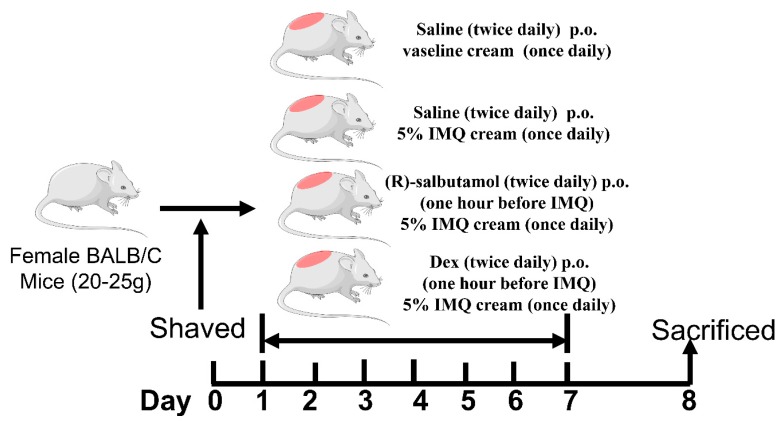
Experimental procedure of the antipsoriatic activity evaluation of (*R*)-salbutamol or Dex. one hour after the administration of different doses of (*R*)-salbutamol or Dex twice per day, mice in all groups except for the control group received a daily topical dose of 62.50 mg of the imiquimod (IMQ) cream on the shaved area of their backs for seven consecutive days. On day 8, the mice were killed to harvest specimens for experiments.

**Figure 2 cells-09-00511-f002:**
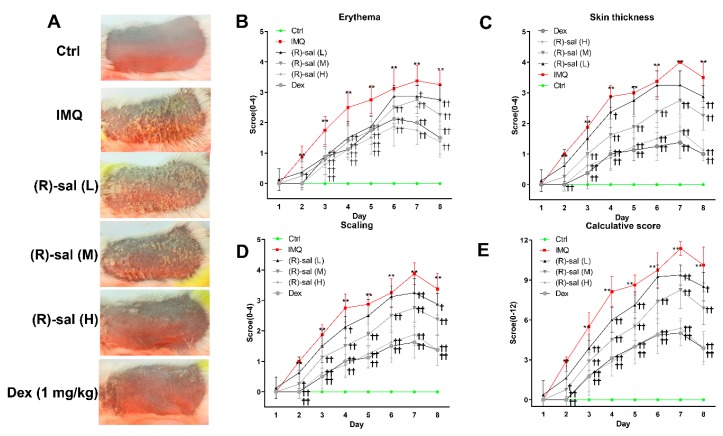
(*R*)-salbutamol alleviates psoriatic dermatitis. Phenotypical presentation of mouse back skin from control, IMQ, (*R*)-salbutamol and Dex groups after seven days of treatment, respectively (**A**). Distinct levels of erythema (**B**), skin thickeness (**C**), scaling (**D**) of back skin was scored daily on a scale from 0 to 4. Additionally, the cumulative score (**E**) (erythema plus scaling plus thickness) is depicted. *n* = 8, Mean ± SD, ^†^
*p* < 0.05, ^††^
*p* < 0.01 vs. IMQ group, ** *p* < 0.01 vs. control group.

**Figure 3 cells-09-00511-f003:**
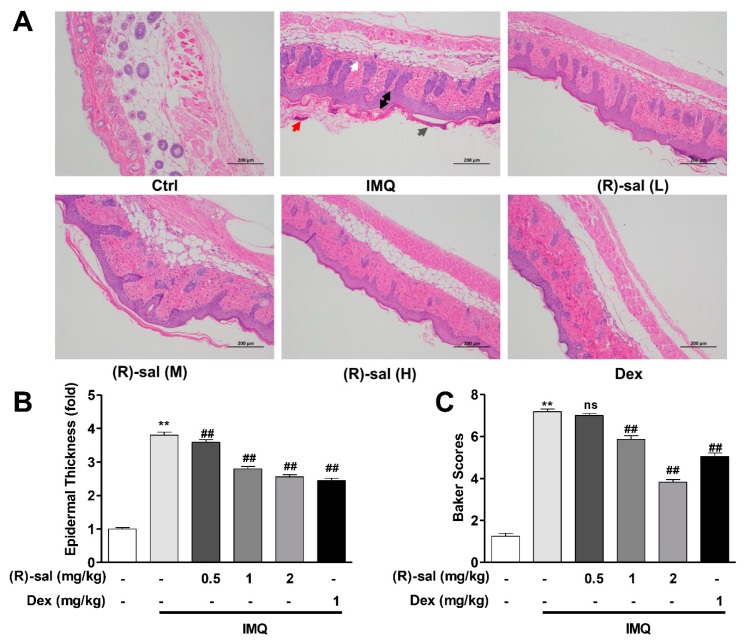
Treatment with (*R*)-salbutamol ameliorated the morphological changes induced by IMQ. (**A**) (*R*)-salbutamol improves pathological injury. (Scale bar: 200 µm) White arrows show inflammatory cell infiltration, gray arrows show parakeratosis, red arrows represent Munro’s microabscesses, and black arrows was thickened prickle cell layer of the epidermis. (**B**) (*R*)-salbutamol alleviates epidermal thickness of the dorsal skin on day 8. (**C**) (*R*)-salbutamol decreased Baker score. ^#^
*p* < 0.05, ^##^
*p* < 0.01, compared with IMQ group. ** *p* < 0.01, compared with control group. Each bar represents the mean ± SD (*n* = 8).

**Figure 4 cells-09-00511-f004:**
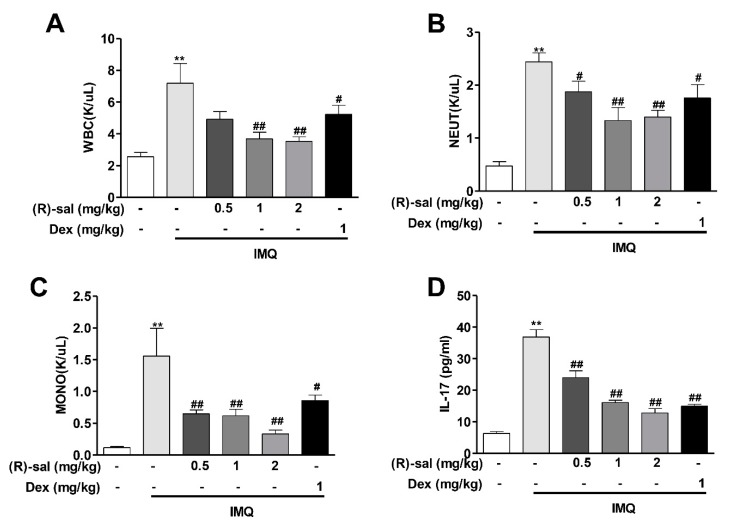
(*R*)-salbutamol reduced the levels of leukocytes in the blood and reduced IL-17 in the plasma. (**A**) white blood cells (WBC), (**B**) Neutrophil, (**C**) Monocyte were analyzed using IDEXX ProCyte DX hematology analyzer. (**D**) Levels of IL-17 in mouse plasma were measured by ELISA. ^#^
*p* < 0.05, ^##^
*p* < 0.01, compared with IMQ group. ** *p* < 0.01, compared with control group. Each bar represents the mean ± SD (*n* = 8).

**Figure 5 cells-09-00511-f005:**
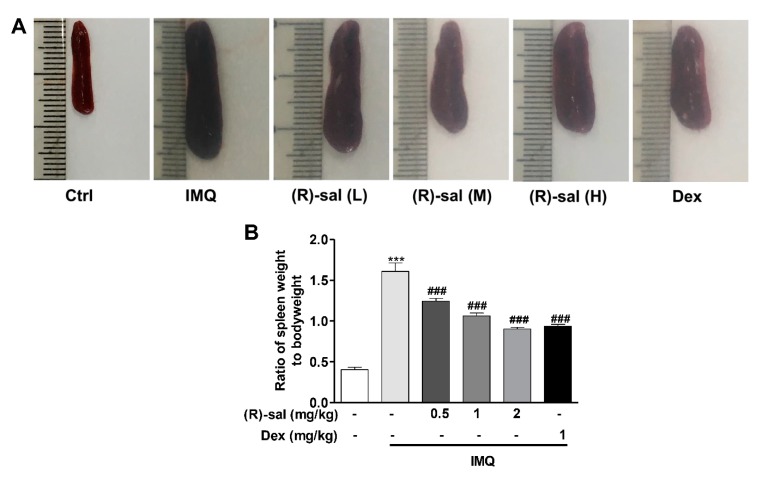
Effect of (*R*)-salbutamol treatment on the ratio of spleen weight to bodyweight. (**A**) Representative photographs of spleen in different groups. (**B**) 24 h after the final administration, mice were sacrificed and the ratio of spleen weight to bodyweight was determined. ^###^
*p* < 0.01, compared with IMQ group. *** *p* < 0.001, compared with control group. Each bar represents the mean ± SD (*n* = 8).

**Figure 6 cells-09-00511-f006:**
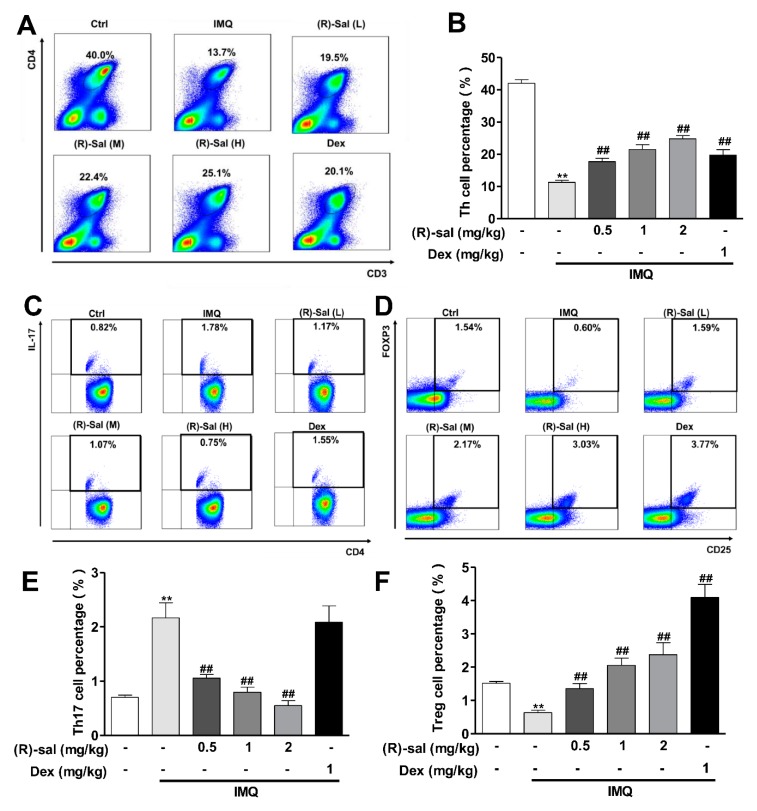
The influence of (*R*)-salbutamol on Th17 cells and Treg cells levels. Spleen cells were obtained from mice on day 8 and then stimulated with cocktail (with Brefeldin) for 6 h. Thereafter, they were stained with fluorescent conjugated anti-mouse CD3, CD25, and CD4. In addition, intracellular staining of IL-17 and Foxp3 was performed using the respective antibodies. Representative contour plots showed the frequency of live CD4+ T cells, IL-17+ Th17 gated and CD25+ Foxp3+ Treg in the splenocytes isolated from mice treated with IMQ and then with (*R*)-salbutamol. Relative scatter plots showed the frequencies determined from live cells. (**A**) Representative dot plots showing the percentage of CD3+CD4+ T cells. (**B**) Statistical analysis of the percentage of CD3+CD4+ T cells. (**C**) Expression of intracellular cytokines IL-17 was detected by flow cytometry in cells gated for CD4+. (**D**) FoxP3 stained with a Foxp3 staining buffer set without stimulation, with CD25+ surface as the gate. (**E**,**F**) Statistical analysis of the above results. ^##^
*p* < 0.01, relative to IMQ group. ** *p* < 0.01, relative to the control group. Error bars represent the mean ± SD (*n* = 8).

**Figure 7 cells-09-00511-f007:**
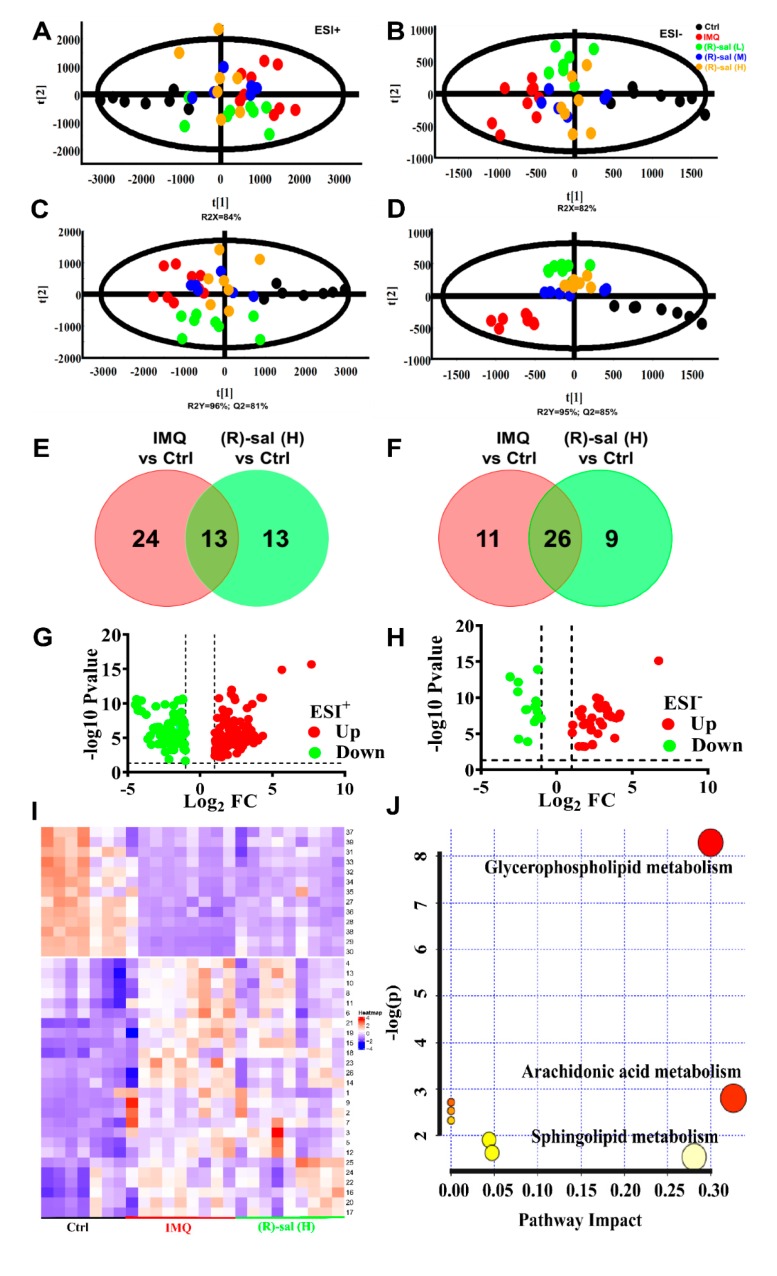
Results of the metabolic effects of (*R*)-salbutamol in mice treated with IMQ to induce psoriasis. (**A**,**B**) PCA plot scores for the control, IMQ and (*R*)-salbutamol (L, M, H) groups in (**B**) ESI (−) mode and (A) ESI (+). (**C**,**D**) PLS-DA score plot for the (*R*)-salbutamol (L, M, H), IMQ and control on the basis of mice plasma metabolic profiles for the (**D**) ESI (−) mode and (C) ESI (+). (**E**,**F**) Venn diagrams showing the upregulated (**E**) or downregulated metabolites (**F**) based on the binary comparison of (*R*)-salbutamol vs. control, IMQ vs. control corresponding to the numbers shown in Appendix A. (**G**,**H**) Volcano plots of *p* values in the (**G**) ESI (+) and (H) ESI (−) mode. (**I**) Visualization of candidate biomarkers among the (*R*)-salbutamol, IMQ, and control in the ESI (+) and ESI (−) mode using Heat map of unsupervised hierarchical clustering. Columns: samples; Rows: biomarkers. The content level of metabolites is denoted by the color key. Red stands for high metabolite level whereas blue color denotes low metabolite level. (**J**) Pathway analysis for the differential metabolism in the (*R*)-salbutamol (L, M, H), IMQ, and control groups based on the topology analysis (*x*-axis) and enrichment analysis scores (*y*-axis). The size and color of each circle represent the pathway impact factor and *p*-value, respectively. The pathways marked in red are the most significant. These analyses were performed using the MetaboAnalyst 4.0 tool.

**Figure 8 cells-09-00511-f008:**
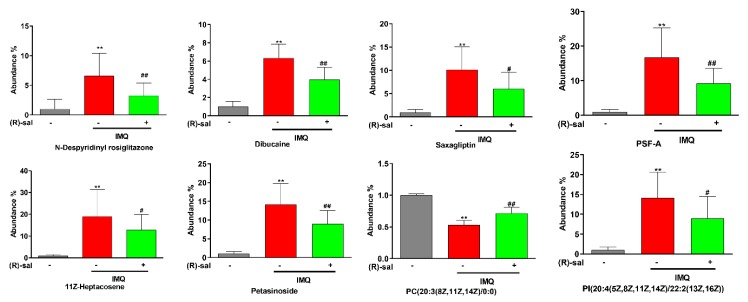
Abundant of some metabolites in plasma of mice from control, IMQ and (*R*)-salbutamol groups. ^#^
*p* < 0.05, ^##^
*p* < 0.01, relative to IMQ group. ** *p* < 0.01, relative to the control group.

**Figure 9 cells-09-00511-f009:**
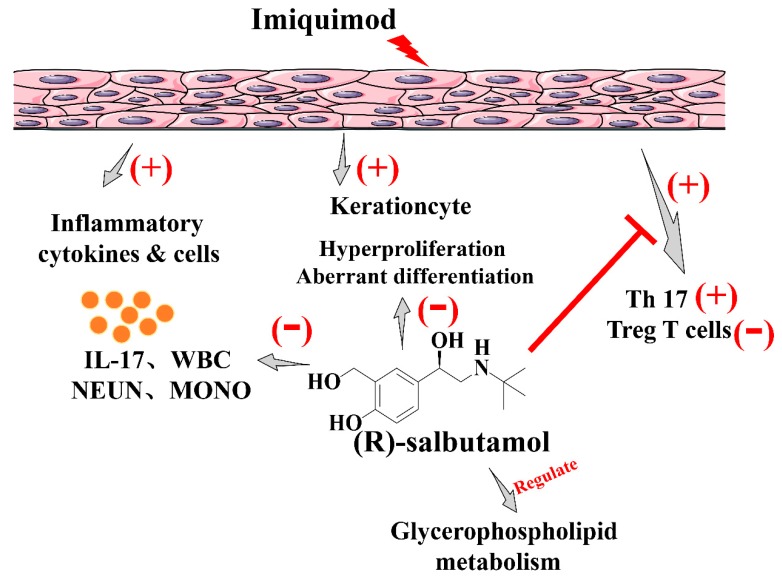
Schematic diagram showing possible mechanisms responsible for the pharmacological efficacy of (*R*)-salbutamol. Oral administration of (*R*)-salbutamol markedly reduced the plasma levels of IL-17, decreased the proportion of CD4+ Th17+ T cells (Th17) whereas increased the percentage of CD25+ Foxp3+ regulatory T cells (Tregs) in the spleens, and affected glycerophospholipid metabolism.

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
