# Peer review of "(R)-Salbutamol Improves Imiquimod-Induced Psoriasis-Like Skin Dermatitis by Regulating the Th17/Tregs Balance and Glycerophospholipid Metabolism"

_cells, 2020, doi:10.3390/cells9020511_

Round 1
Reviewer 1 Report
Dear Authors,
Results presented in this manuscript are interesting.
Comments:
Lines 116 to 121: The text is a little repetitive (line 116 vs 119-121). Please, describe what “MACS” is and also what the cocktail with Brefeldin A is for, since it is the “MACS” that blocks splenocytes.
Treatment methodology, Figure 1 and Figure 1 legend: Some details about the given treatment are not clear. Does the mice were treated with IMQ prior to the treatment with (R)-salbutamol? If yes, please include the pre-treatment in the figure as well. This is not clear if the IMQ group only received a once daily IMQ treatment or if it's all the IMQ-induced mice (treated with (R)-sabutamol). There was no pre-treatment required for the induction of IMQ-induced mice before the treatment with (R)-sabutamol?
Figure 1 legend: (Line 175) Antipsoriatic activity
Figure 2 Legend: Why are there two titles to the figure legend? (R)-salbutamol alleviates psoriatic dermatitis. (R)-salbutamol antagonizes the effects of IMQ. It is a little repetitive.
Figure 2 Legend: Please change “skin thickening” in the legend for “skin thickness” in order to be consistent with the figure and the legend.
In figure 2: How does the skin thickness was measured without any biopsies and sacrificing animals each day?
In figure 2: The use of hashtag makes it difficult to understand the graph. Is it possible to change for crosses (†)?
Figure 3: It would be easier to compare the different conditions if the skin sections were aligned on the same side (example: epidermis on top of all pictures). Please adjust so that everything is in the same direction. Would it be possible to indicate the different layers of the skin (dermis/epidermis)? This would allow a better comparison of the thicknesses thereafter. Do you have a better picture for the histological analyses of Dex. The chosen section is not of good quality.
What is the difference between the epidermal thickness of figure 2 and figure 3?
Lines 187-188: It is written that HE examination showed the high inflammatory cell infiltration, acanthosis with extended rete ridges, Munro's microabscesses, parakeratosis, high number of prickle cells, and thickened prickle cell layer of the Can you please indicate these features in your Figure 3 with arrows? Personally, I think that the size of the picture does not allow the analyses of inflammatory cell infiltration.
Lines 203-206: We observed that IMQ group had significantly increased total number of WBC, NEUT and MONO (P < 0.01) compared to the control group. Importantly, a subsequent oral administration of different doses of (R)-salbutamol or Dex lowered the increased number of WBC, NEUT and MONO, implicating that (R)-salbutamol suppresses inflammatory responses by decreasing the number of WBC, NEUT and MONO in the blood (Figure 4A, B and C).
Line 215: Compared with the control group, the level of IL-17 in the model group was increased by 5.82 fold. Please precise which group is the model group.
Line 220: The ratio of spleen weight to body weight was significantly decreased in the (R)-salbutamol or Dex treatment group, even not to the level of the ratio of the control group. Revise the structure of this sentence.
Figure 6: How did you determine the cell viability? Why did you analyze immune cells of the spleen and not immune cells infiltrated in the lesional skin or blood?
Please revise figure 7 legend, which is confusing.
Figure 7. Results of the metabolic effects of (R)-salbutamol in mice treated with IMQ to induce psoriasis. (A, B) PCA plot scores for the control, IMQ and (R)-salbutamol (L, M, H) groups in (B) ESI (-) mode and (A) ESI (+). (C, D) PLS-DA score plot for the(R)-salbutamol (L, M, H), IMQ and control on the basis of mice plasma metabolic profiles for the (D) ESI (-) mode and (C) ESI (+). (E, F) Volcano plots of p values in the (E) ESI (+) and (F) ESI (-) mode. (G, H) Venn diagrams showing the upregulated (G) or downregulated metabolites (H) based on the binary comparison of (R)-salbutamol vs. control, IMQ vs control corresponding to the numbers shown in Supplemental Table S4. (I) Visualization of candidate biomarkers among the (R)-salbutamol, IMQ, and control in the ESI (+) and ESI (-) mode using Heat map of unsupervised hierarchical clustering. Rows: samples: columns: biomarkers. The content level of metabolites is denoted by the color key. Red stands for high metabolite level whereas blue color denotes low metabolite level. (J) Pathway analysis for the differential metabolism in the (R)-salbutamol (L, M, H), IMQ, and control groups based on the topology analysis (x-axis) and enrichment analysis scores (y-axis). The size and color of each circle represent the pathway impact factor and p-value, respectively. The pathways marked in red are the most significant. These analyses were performed using the MetaboAnalyst 4.0 tool.
The Venn diagrams are not (G and H) but E and F. Please adjust with the rest of the associated legend (E = upregulated and F = downregulated metabolites?). I) Rows = samples? Are the columns really the biomarkers or rather the different mice. Please review the legend and make sure that everything is ok. I) Identify the metabolites presented in the Heatmap. The identification must be on the figure, not in the supplementary Table. If there is not enough space put some metabolites not discussed in the paper in supplementary materials.
Lines 360, 385, 389: It looks like there are some missing references (#48, 54, 53 and 55, respectively).
Line 385: Revise structure of the following sentence. The downregulation of glycerophospholipids may because of cell membrane damage.
Line 392: Several studies have stressed… Only one study is cited. Please add references.
Line 395-398: The b-adrenergic receptor is implicated in the regulation of adenylate cyclase and production of cAMP. It regulates cell proliferation and differentiation. Thus, changes in the metabolism of sphingolipids could be caused by the restoration of epidermal differentiation following the treatment. Please explain further the link between the regulation of Th17/Tregs by sphingolipids in this study. Maybe looking at one particular deregulated sphingolipid would be interesting.
Is treatment targeting the beta-adrenergic receptor adequate to treat psoriasis, since the beta-adrenergic receptor is impaired in psoriasis?
Discussion: It is slightly strong to say that the (R)-salbutamol attenuates the psoriatic phenotype by regulating Th17 and Treg with the differences obtained with flow cytometry. Can you discuss how the small decrease in Th17 (1%) and the increase in Treg observed can support this point?
Author Response
Please see the attachment, Thanks

Reviewer 2 Report
The present study assessed the effects of salbutamol on a murine imiquimod (IMQ) mediated psoriasiform model. The investigators elegantly showed that the b agonist is capable of inhibiting all the clinical parameters associated with the psoriasiform effects of IMQ. To this effect, they presented Figure 2 with graphs and photographs. The latter are of very poor quality and should be replaced by better ones, or otherwise remove them entirely.
Figure 4 shows the effects on circulating leukocytes: this is interesting because the topical application of IMQ results in neutrophilia, a finding that has not been reported in the literature. Furthermore, dexamethasone reduces the number of circulating neutrophils induced by IMQ. This needs some comment since the systemic effects of glucocorticosteroids are known to induce marked increases in the number of neutrophils in the circulation. How do the authors explain their result which is counterintuitive?
Of much interest are the findings that salbutamol acts through its effects on lymphocytes, reducing IL-17 in addition to the metabolomic studies supporting effects on glycerophospholipid metabolism.
In conclusion, the manuscript provides a solid argument for the future trial of salbutamol as a therapy for psoriasis.
Author Response
Please see the attachment, Thanks.

Reviewer 3 Report
I found the reaserch in the manuscript interesting, but it needs improvment. Although the English is correct, the Flow of the writting is very dense, and I found it difficult to read. The descriptioin of the methods and statitical analysis must be improvedAuthor Response
Please see the attachment, Thanks.
